# Traction Therapy for Cervical Radicular Syndrome is Statistically Significant but not Clinically Relevant for Pain Relief. A Systematic Literature Review with Meta-Analysis and Trial Sequential Analysis

**DOI:** 10.3390/jcm9113389

**Published:** 2020-10-22

**Authors:** Claudio Colombo, Stefano Salvioli, Silvia Gianola, Greta Castellini, Marco Testa

**Affiliations:** 1Department of Neuroscience, Rehabilitation, Ophthalmology, Genetics and Maternal-Infantile Sciences, University of Genoa, 16121 Genoa, Italy; colombo.claudio92@gmail.com (C.C.); stefano.salvioli@gmail.com (S.S.); marco.testa@unige.it (M.T.); 2Unit of Clinical Epidemiology, IRCCS Istituto Ortopedico Galeazzi, 20162 Milan, Italy; greta.castellini@grupposandonato.it

**Keywords:** cervical radiculopathies, traction, meta-analysis, systematic review, trial sequential analysis, evidence-based practice

## Abstract

Aim: We aimed to investigate the effectiveness of traction therapy in reducing pain by performing a systematic review with meta-analysis. We also explore the best modality for administering traction to patients with cervical radicular syndrome (CRS). Methods: We searched the Medline, Physiotherapy Evidence Database (PEDro), Cochrane Central Register of Controlled Trials, and Cumulative Index to Nursing and Allied Health Literature (CINAHL) electronic databases. Two reviewers independently selected randomized controlled trials (RCTs) that compared traction in addition to other treatments versus the effectiveness of other treatments alone for pain outcome. We calculated the mean differences (MDs) and 95% confidence intervals (CIs). We used Cochrane’s tool to assess risk of bias and the Grading of Recommendations Assessment, Development and Evaluation (GRADE) system to evaluate the quality of evidence and summarize the study conclusions. Results: A total of seven studies (589 patients), one with low risk of bias, were evaluated. An overall estimate of treatment modalities showed low evidence that adding traction to other treatments is statistically significant (MD −5.93 [95% CI, −11.81 to −0.04] *P* = 0.05 and I^2^ = 57%) compared to other treatments alone. The subgroup analyses were still statistically significant only for mechanical and continuous modalities. Conclusions: Overall analysis showed that, compared to controls, reduction in pain intensity after traction therapy was achieved in patients with cervical radiculopathy. However, the quality of evidence was generally low and none of these effects were clinically meaningful.

## 1. Introduction

Cervical radicular syndrome (CRS) is characterized by pain symptoms radiating to one or both arms in a dermatomeric distribution of sensorimotor symptoms due to a conduction deficit and a dynatomeric distribution of pain [1,2,3]. CRS occurs following radicular inflammation or compression of the cervical spine. The annual incidence is 83.2 per 100,000, with an increase in prevalence through the fifth decade of life [1,4,5]. A more recent study from the US military found an incidence of 1.79 per 1000 person-years [6].

There is no clearly established consensus regarding indications for CRS treatments. A common approach recommends surgical treatment when there are several neurological deficits, when there is a strong correlation between imaging and clinical signs and symptoms or when there is a failure or relapse of symptoms after six weeks of conservative treatment [7]. Guidelines suggest surgery for the rapid relief of pain in CRS in short terms [8] when radicular pain is caused by degenerative cervical spondylosis. Nonetheless, evidence is insufficient for defining the optimal time for surgery [6] and the best surgical approach [7,9]. In all other conditions, conservative treatment should be preferred. In fact, patients undergoing conservative treatment have reported good to excellent outcomes in up to 90% of cases [10]. Conservative treatment of CRS has included different interventions such as manual therapy (mobilization and manipulation), physical therapy and traction [6]. Physiotherapy and manual therapy seem to be effective in reducing pain in patients with CRS [1,11]. Traction treatment has been widely investigated in physiotherapy [1,4,11,12,13,14,15,16,17,18,19,20,21,22,23,24,25,26,27,28], and clinical prediction rules for the successful application of the technique have been developed [29]. Traction can be administered with different modality: mechanical or manual and in an intermittent or continuous mode [30,31], with the aim to increase intervertebral space, inhibit nociceptive impulses, and reduce spinal nerve root compression [22]. More recent studies have found that traction can halt self-maintenance of pain by stimulating proprioceptive receptors [32]. Despite its wide and renowned use, the evidence behind sustained traction efficacy in CRS is unclear. Systematic literature reviews have produced conflicting evidence for the effectiveness of traction in treating CRS. Graham et al. [30,31] performed two reviews on the efficacy of cervical traction by setting a different question framework. The authors included a wide variety of neck disorders (e.g., headache, whiplash, mechanical neck disorders). Differently, two other reviews involved only CRS patients [1,33]. In 2013, Thoomes et al. [1] focused on the effectiveness of different conservative treatments for patients with CRS, and concluded with minimal evidence that traction was no more effective than the control. However, the review was limited to three trials. Later, in 2018, Romeo et al. [33] found some support for the use of the mechanical and manual traction for CRS in addition to other physical therapy procedures for pain reduction. However, they performed only a subgroup analysis for mechanical and manual modalities.

Traction is a passive therapy modality that can be compared to a spontaneous course (sham or no therapy), or to other passive or active interventions (e.g., exercise). To date, there have been no high-quality studies that have demonstrated the spontaneous course of cervical radiculopathy [6]. Firstly, to assess the effectiveness of traction therapy, it is important to confirm the superiority of traction against a spontaneous course. The aim of this review was to investigate the effectiveness of any traction technique (alone or in addition to other interventions) versus sham or no therapy, and to apply subgroup analysis in order to explore the best mode of treatment delivery (manual/mechanical and continuous/intermittent). These two aspects have never been clarified by previous reviews. In order to settle any related uncertainty, we also performed trial sequential analysis (TSA). Finally, we offer an updated review adding new trials and provide a risk of bias (RoB) assessment and judgment of the quality based on the GRADE method [34,35,36].

## 2. Methods

### 2.1. Protocol and Registration

The review protocol was registered at the International Register of Systematic Reviews with the following code CRD42017081089, available at http://www.crd.york.ac.uk/PROSPERO/display_record.php?ID=CRD42017081089.

### 2.2. Search Strategies and Selection Criteria

We searched the Medline, PEDro, Cochrane Central Register of Controlled Trials, and CINAHL databases up to May 2020. In Appendix B we reported the Medline full search strategy, adapted for all other databases. We also hand-searched bibliographies of the relevant studies, systematic reviews, the register of randomized controlled trials (RCTs; e.g., clinicaltrials.gov), and groups of interest. Study selection was carried out by two independent authors and disagreements were solved by a third author.

Based on our PICO question (population, intervention, comparison and outcomes), eligible studies were selected if they were RCTs that included adults (>18 years) with neck pain radiating to one or both arms associated with sensorimotor dysfunction, as well as a diagnosis of CRS established by clinical examination or diagnostic imaging or both. The following comparisons were investigated: (**i**) traction versus sham traction or no treatment; (**ii**) traction in addition to other intervention (e.g., physiotherapy) versus other intervention. We excluded RCTs investigating a head-to-head comparison (traction versus other intervention). The primary outcome was pain intensity, as assessed by any measurement (e.g., visual analogue scale (VAS)) at the short-term (after the end of a series of treatments over three to four weeks). Only RCTs without a limitation on year of publication in English were included.

### 2.3. Risk of Bias of Included Studies

Risk of bias was assessed independently by two authors using Cochrane’s Risk of Bias Tool [37,38]. Since the pain outcome assessment was patient-reported, we rated allocation concealment, blinding of participants, and outcome assessors as essential criteria for overall judgment. When all three procedures were adequate, risk of bias was graded as low. Otherwise, if only one of these was inadequate, risk of bias was graded high. Risk of bias was judged unclear in other cases. A third author was consulted to solve eventual disagreements in the judgments.

### 2.4. Data Extraction

Two independent authors extracted trial characteristics (e.g., study design, country of contact author, description of treatment and control, pain outcome measure) patient characteristics (e.g., description of CRS diagnosis, sample size, and drop-outs) and outcome scores. Disagreements were solved by consultation with a third author.

### 2.5. Data Synthesis

We used mean differences to assess the treatment effects of continuous outcomes: we expected similar scales with different ranges that measured pain intensity (e.g., VAS 0–10 and VAS 0–100), which we then linearly transformed into a 0–100 scale (correcting the SDs accordingly) in order to clinically compare the pain scales. We used post-treatment values instead of mean change because we assumed similarity between groups at baseline and because we noted that mean change values were frequently not reported [39]. In order to include multiple intervention arms and overcome a unit of analysis error, we split the “shared” group into two or more groups of smaller sample sizes and included two or more comparisons, as suggested by the Cochrane Collaboration [40]. The corresponding authors of the trials were contacted to obtain missing information.

Data were meta-analyzed using random effect models [41]. Heterogeneity was assessed and interpreted using I^2^ and chi-squared tests, as proposed by Higgins et al. [42]. If considerable heterogeneity (I^2^ > 70%) was detected, a sensitivity analysis for risk of bias assessment was run. Moreover, we performed subgroup analysis for each therapy modality in order to capture any effect modifiers. The meta-analyses were performed using RevMan Software [43].

### 2.6. Trial Sequential Analysis

We applied Trial Sequential Analysis (TSA) to evaluate the robustness of evidence and estimate the diversity-adjusted required information size (DARIS) (i.e., the number of participants needed in a meta-analysis to detect or reject a certain intervention effect) and to control for random errors with trial sequential monitoring boundaries [44,45]. DARIS was calculated based on: (i) a predefined anticipated intervention effect equal to a minimal important difference (MID) between groups of 20 points on the pain outcome measured on a standardized scale of 0–100 mm [46], (ii) a standard deviation observed in the control group of trials with a low risk of bias, (iii) an alpha of 5%, (iv) a beta of 20%, and (v) an assumed diversity of 71% calculated among the trials in the meta-analysis (variance-based model). The analysis was performed using TSA software beta version 0.9.5.10 [47].

### 2.7. Quality of Evidence and Summary of Findings

The quality of evidence was assessed using the GRADE approach [48] and GRADEpro software [34]. Assessment was based on five factors: risk of bias in study design, heterogeneity of results, indirectness, imprecision, and publication bias [35,36,49]. Two authors independently conducted the assessment; disagreements were solved by the third author if consensus was not met.

## 3. Results

### 3.1. Study Selection

Seven RCTs of the 81 studies screened as eligible were included in this systematic review but only six were used for meta-analysis. The study selection process and reasons for exclusion are presented in Figure 1.

### 3.2. Study Characteristics

Table 1 presents the study characteristics. Overall, the seven trials involved a total of 589 patients recruited from a variety of settings (outpatient clinics, hospitals, rehabilitation clinics and universities) according to different diagnostic criteria. For details of interventions see Appendix A.

The most frequently reported technique was mechanical traction (*n* = 5.71%) [12,13,14,16,22] in continuous mode (*n* = 5.71%) [12,13,14,16,18]. The median follow-up period was four weeks. Traction was administered on average for 10–25 min, at an angle of 15–25° of cervical flexion or in the most pain-free position, with a load of 2.7–15.91 kg. Overall, we obtained seven comparisons of cervical traction in addition to other interventions versus the other interventions one their own [12,13,16,18,28], including one of cervical traction in addition to other interventions against the latter associated with sham traction [14], and one comparison of cervical traction against sham traction [22]. No study compared cervical traction with no treatment at all. However, the study by Moustafa et al. compared cervical traction and other interventions to other interventions alone via an innovative methodology based on H-reflex biofeedback [16]; this study was excluded from meta-analysis and TSA in the present review. We also followed the template for intervention description and replication (TIDieR) checklist for reporting important descriptions of traction in RCTs.

### 3.3. Risk of Bias

The risk of bias of RCTs is presented in Figure 2. Overall, one RCT was judged as having an unclear risk of bias [22], one as having a low risk of bias [14], and five as having a high risk of bias [12,13,16,18]. Risk of bias due to lack of blinding was predominant (*n* = 5 RCTs); the blinding procedure was considered adequate in two studies. All seven studies were described as randomized, although only three reported appropriate methods for sequence generation and allocation. Four had an acceptable drop-out rate and two were consistent in outcome reporting.

## 4. Synthesis of Results

### 4.1. Traction Versus Sham Traction

Only one study [22] was found that compared continuous mechanical traction against sham traction. With a low quality of evidence (Appendix A), the effect of traction compared to the control was not significantly different (Figure 3: MD −5.00 [95% CI, −14.98, 4.98] *P* = 0.33 > 0.05).

### 4.2. Traction in Addition to Other Treatments Versus Other Treatments

Six studies [12,13,14,16,18,28] with eight informative comparisons (involving 405 patients) measured pain intensity. With a low quality of evidence (Appendix A), the effect of traction in addition to other treatments was significantly different compared to the control (Figure 4: MD −5.93 [95% CI, −11.81 to −0.04] *P* = 0.05 and I^2^ = 57%). Mechanical traction used in six of the eight comparisons and manual traction was used in the remaining two.

In TSA (Figure 5), the cumulative z-curve crossed both the conventional threshold of 0.05 for statistical significance and the monitoring boundary for superiority (i.e., benefit), showing a beneficial effect of traction in addition to other treatments versus other treatments as the control for reducing pain in CRS (MD −5.93 [95% CI, −11.81 to −0.04].

## 5. Additional Analysis

### 5.1. Subgroup Analysis

We performed a subgroup analysis for the mode of treatment delivery in the following comparison: traction + other treatments compared to other treatments.

### 5.2. Mechanical and Manual Traction

Mechanical traction was assessed in five studies (total of 357 patients) [12,13,14,16,18]. With a low quality of evidence the subgroup analysis of mechanical traction showed a significant statistical difference between intervention (i.e., traction in addition to other treatments) versus control (i.e., other treatments) (Figure 6a; MD −6.21 [95% CI, −11.69 to −0.73] *P* = 0.03 < 0.05 and I^2^ = 36%, Appendix A). Mechanical traction was administered with the patient supine [12,13,14,16,18] in five comparisons and in a sitting position in one. Through those comparisons, traction was applied intermittently in four groups [12,14,16,18] and continuously in the other two groups [12,13]. The duration of traction was 15–25 min on average at an angle of 15–25° of cervical flexion, or in the most pain-free position, with a load of 2.7–15.91 kg.

Manual traction was assessed in two studies (total of 48 patients) [13,28]. With very low quality evidence, no differences between intervention and control were found in manual traction (Figure 6a; MD −9.26 [95% CI, −38.54 to 20.03] *P* = 0.54 and I^2^ = 85%, Appendix A). Through those comparisons, intermittent traction was applied as follows: 20 repetitions × 20 s; 6 kg load in the most pain free position in one study and for 10 min (10 s tension/5 s relaxation) at 25° cervical flexion in the other.

### 5.3. Continuous Traction and Intermittent Traction

With very low quality evidence, two studies (involving 60 patients) [12,13] performing continuous traction reported a statistically significant difference between intervention and the control (Figure 6b; MD −13.08 [95% CI, −24.29 to −1.88] *P* = 0.02 < 0.05 and I^2^ = 0%, Appendix A). The duration of continuous traction was 15–25 min on average at an angle of 15–25° of cervical flexion or in the most pain-free position, with a load of 2.7–15.91 kg. The effectiveness of intermittent traction was evaluated in six studies (involving 345 patients) [12,13,14,16,18,28]. With very low quality evidence, a statistically non-significant difference between intervention and control was found (Figure 6b; MD −4.27 [95% CI, −10.67 to 2.12] *P* = 0.19 > 0.05 and I^2^ = 60%, Appendix A). The duration of traction was 10–20 min on average at an angle of 15–25° of cervical flexion or in the most pain-free position. The load (range 5–15.91 kg) was applied in a tensioning stage (about 10–60 s in duration) and a relaxation stage (5–20 s in duration).

## 6. Discussion

Traction seems to be superior to other conservative treatments individually when combined with them, albeit with a low quality of evidence and the need for caution in interpreting the effects. Looking at the explorative subgroup analysis, some indications for the modality of delivery can be suggested in favor of mechanical traction and continuous delivery modalities. In fact, it seems that with solid consistency, mechanical traction is still statistically significant as opposed to manual traction, and continuous traction is consistently significant (while intermittent traction is not). Nevertheless, the difference between subgroups is not significant. Maybe the overall statistical significance can benefit more from the greater representation of mechanical and continuous comparisons. Among the several systematic reviews investigating the effectiveness of conservative approaches and physiotherapy in the treatment of CRS, most have reported qualitative synthesis [4,7,23,24,25,27]. Some meta-analyses focusing on the effectiveness of traction have reported discrepant conclusions. For example, Graham et al. [31] concluded in their analysis that the literature did not support or refute the efficacy or effectiveness of continuous or intermittent traction for pain reduction. Of note, Graham included patients with neck pain, only some case of which were associated with radiculopathy, which reduced the generalizability of the results. Thoomes et al. [1] found that cervical traction was not effective in the treatment of CRS, whereas Romeo et al. [33] reported an overall statistically significant effect of combining traction with other conservative treatments versus other conservative treatments individually.

Our analysis updates the evidence with a new RCT that has never been included in meta-analysis [28]. We offered an interpretation of results balancing the TSA, risk of bias assessment and the GRADE approach, which enabled us to make a more transparent and structured judgment based on criteria for confidence of the quality of evidence [35]. TSA is a powerful method that can control the risk of random errors and show whether a sufficient amount of information has been reached to draw a conclusion, and it can estimate how much more information is still needed to adopt or discard an intervention effect. However, because TSA does not consider systematic errors, GRADE assessment helps to strike a balance between the magnitude of effects and the quality of evidence. Using GRADE, we were able to assign a low level of quality of evidence to the traction + other treatments versus other treatments individually, chiefly due to the low internal validity of the studies. Assuming that blinding is not always possible in non-pharmacological interventions [50], we believe that the use of a sham traction technique as a control group can better minimize performance bias in this setting, thus making the intervention indistinguishable from the control. Yet we have found that very few studies have compared traction with sham traction. Furthermore, outcome assessment should be measured by an external assessor independent of the study to minimize the influence of the investigators involved in the trial, especially in subjective outcome assessments such as pain [37].

The TSA with a standard value for type I and II errors (i.e., alpha and beta) confirmed the effectiveness of any traction technique added to other treatments versus other treatments on their own as the control, reaching the optimal information size. All stakeholders should be informed that traction is significantly effective statistically, but without clinical relevance to patients. Our predefined anticipated intervention effect (a MID of 20 points on pain) was not reached [46,49]. This means that integrating or not integrating traction in the treatment of CRS does not significantly modify the perceived effect on pain symptoms.

There are several plausible explanations for the failure to achieve clinical relevance. CRS lacks a precise clinical definition. As a consequence, the inclusion criteria in primary studies are often too broad, non-specific and unclear. For example, including patients with chronic symptoms and/or the use of diagnostic imaging can reduce the homogeneity of a patient’s condition, thus creating a controversial limitation for external validity and quality of evidence on which recommendations can be based. Currently, pain science argues that the inclusion of chronic symptoms (i.e., central sensitization) can be related to a change in the mechanisms that maintain pain (i.e., psychological), confounding the effectiveness of study intervention [51,52,53]. In the same way, the literature has demonstrated the low validity of diagnostic imaging in relation to symptoms, which may be useful only when red flags are present [54]. The use of traction therapy for CRS does not appear to be essential in clinical practice. However, pain measurement is a patient-reported outcome that can be influenced by psychological mechanisms. In the light of these considerations, this treatment option should only be reserved for patients with acute symptoms or for patients that show a reduction of symptoms (i.e., pain) when subjected to cervical traction during the physical examination [5].

Further studies involving patient subgroups are needed to confirm our hypotheses and to gain firm evidence on traction therapy. A future area of focus is other techniques (e.g., thoracic manipulation, neurodynamic sliding, vertebral gliding) for CRS treatment in trials that apply more homogeneous inclusion criteria and adopt explicit methods that minimize selection, performance, and detection of bias.

## 7. Study Limitations

Our review has several limitations. We did not investigate other functional outcomes (e.g., abilities of daily living) or adverse events. Publications were only reviewed in English. Finally, to obtain a broader view of different traction techniques, we thought it necessary to include a wide variety of control groups, which may have reduced the accuracy of comparisons. However, the paucity of trials precluded a further subgroup analysis. We only focused on the superiority of the traction versus the sham or no therapy at all (or traction in addition to other interventions versus other interventions alone). We did not select trials versus other passive/active interventions (e.g., traction versus exercise). Moreover, most trials investigated the addition of traction to other interventions. This increase in the supply of care could potentially have been influenced by a placebo effect [55], even if the randomization of multimodal approaches should care for balanced therapeutic dose in all treatment arms. Either way, multimodal approaches are usually present in pragmatic trials where standard care should be guaranteed.

Since we found a statistical significance in favor of traction therapy, future studies should investigate a head-to-head comparison of active versus passive interventions.

## 8. Conclusions

Meta-analyses revealed a low quality of evidence: Traction seems to be superior to other conservative treatments when combined with those treatments in reducing pain in patients with CRS at a three- to four-week follow-up assessment, but the findings were not clinically relevant. The results of subgroup analyses were statistically significant only for mechanical and continuous traction. Trial sequential analysis confirmed the effectiveness of traction therapy, but caution in interpreting the results is warranted due to a widely adjusted confidence interval and lack of clinical relevance. Further studies are needed to gain firm evidence on this topic. Future RCTs should investigate other interventions for CRS, apply homogeneous and universally accepted inclusion criteria and clinical examinations, focus on patients with acute symptoms and adopt explicit methods to minimize selection, performance and detection bias.

## Figures and Tables

**Figure 1 jcm-09-03389-f001:**
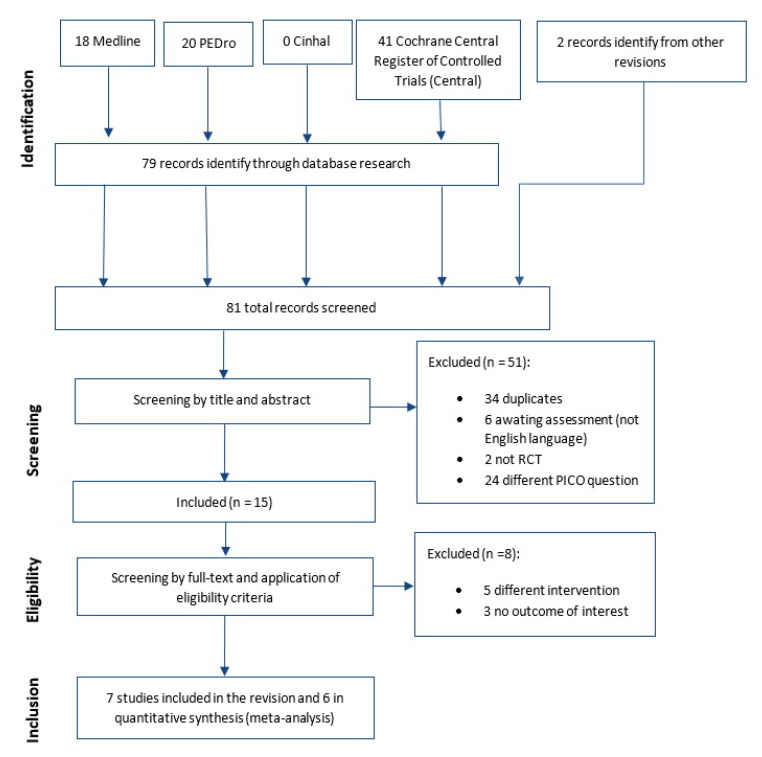
Study flow diagram. PEDro= Physiotherapy Evidence Database; RCT: Randomized controlled trial; PICO= population, intervention, comparison and outcomes.

**Figure 2 jcm-09-03389-f002:**
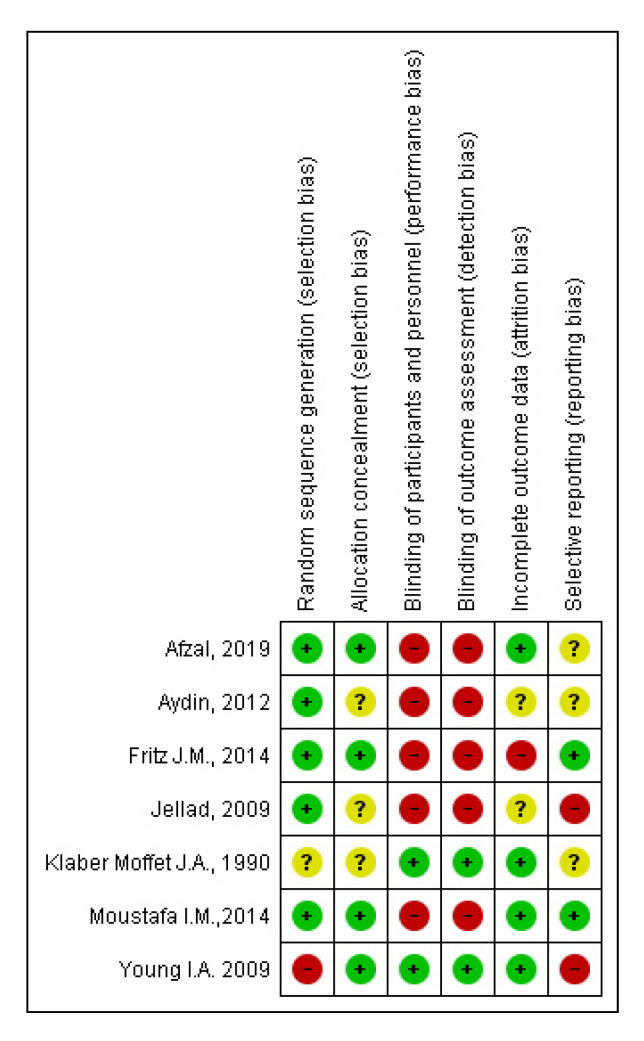
Risk of bias graph. Legend: Red (-) = high risk of bias; Yellow (?) = unknown risk of bias; Green (+) = low risk of bias.

**Figure 3 jcm-09-03389-f003:**
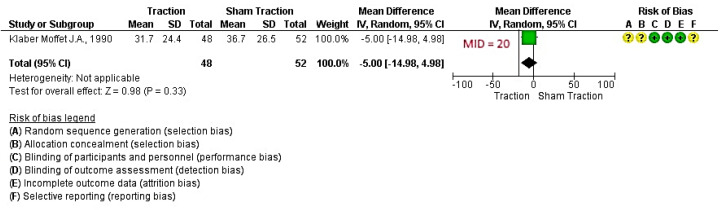
Effect of traction versus sham traction for reducing pain in cervical radicular syndrome. Legend: Red (-) = high risk of bias; Yellow (?) = unknown risk of bias; Green (+) = low risk of bias; MID= Minimal important difference; IV= Inverse of Variance method.

**Figure 4 jcm-09-03389-f004:**
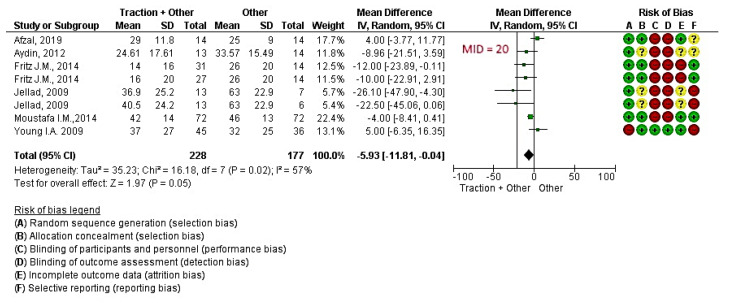
Effect of traction + other treatments versus other treatments for reducing pain in cervical radicular syndrome. Legend: Red (-) = high risk of bias; Yellow (?) = unknown risk of bias; Green (+) = low risk of bias; MID= Minimal important difference; IV= Inverse of Variance method.

**Figure 5 jcm-09-03389-f005:**
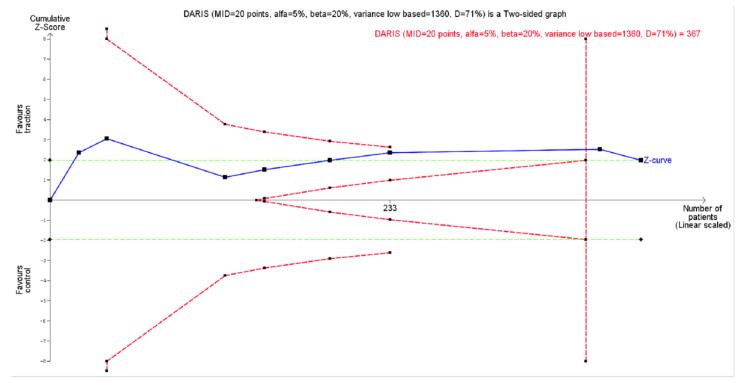
Trial Sequential Analysis (TSA) for pain in traction versus the control in cervical radicular syndrome. Diversity adjusted required information size (DARIS) of 367 patients was calculated using α = 0.05 (two sided), β = 0.20 (power 80%), diversity (D^2^) = 71%, with a minimal important difference (MID) of 20 points out of 100 mm. The blue cumulative z curve was constructed using a random effects model. Solid squares denote individual trials; trials are plotted in chronological order (from left to right). The *x*-axis indicates the cumulative number of patients; the starting point of the *z*-curve is at *x* = 0 (i.e., inclusion of no trials). The dotted red lines indicate the monitoring boundaries for superiority, inferiority or futility, while the dotted green lines indicate the boundaries for a conventional statistic (i.e., 0.05).

**Figure 6 jcm-09-03389-f006:**
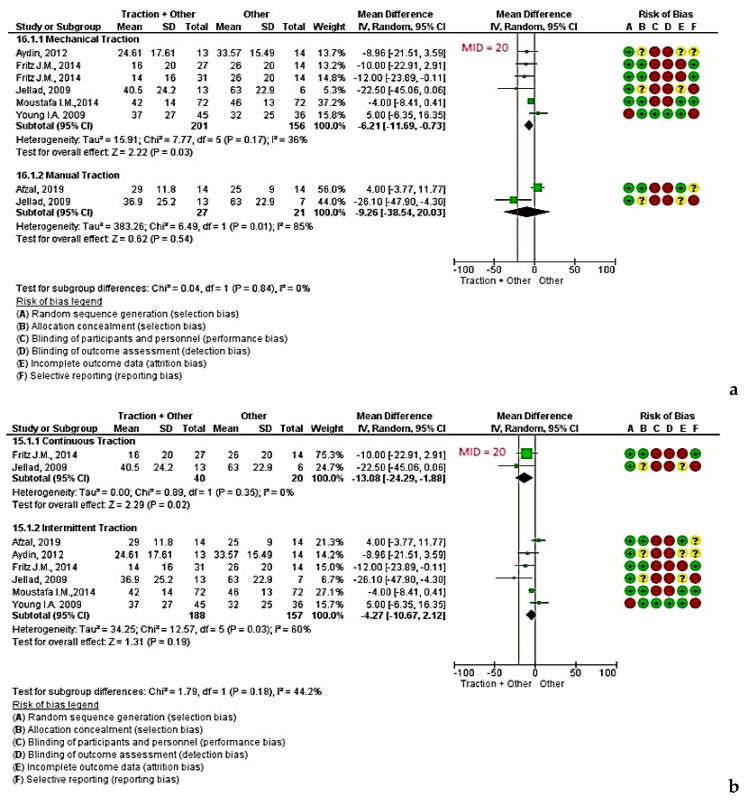
(**a**) Mechanical and manual traction versus control; (**b**) continuous and intermittent traction versus control. Legend: Red (-) = high risk of bias; Yellow (?) = unknown risk of bias; Green (+) = low risk of bias.

**Table 1 jcm-09-03389-t001:** Study characteristics.

Author, Year	Study Design	Country	Diagnostic Criteria	Intervention Group	Control Group	Outcome Measure Considered (Follow Up)	Sample Size (Drop Out)
Afzal R., 2019	RCT	Pakistan	Clinic criteria: unilateral upper extremity pain, paresthesia, or numbness, and at least 3/4 positive test of Wainner’s cluster. Imaging: MR diagnosis of radiculopathy.	I1: Manual opening technique + manual intermittent traction 10 min, 25° cervical flexion, 10 s tension/5 s relaxation. I2: Manual intermittent traction 10 min, 25° cervical flexion, 10 s tension/5 s relaxation.	C: Manual opening technique pull the neck to rotation at the restriction improving rotation and performing opening 3 sets of 10 repetitions	NPRS 0–10 (3 weeks)	40 (1)
Fritz J.M., 2014	RCT	Utah, USA	Clinic criteria: neck pain associated with pain or numbness distally to a/c joint or caudal to superior edge of scapulae.	I1: Physiotherapy + mechanical intermittent traction 15 min, 15° cervical flexion, 60 s tension/20 s relaxation. I2: Physiotherapy + over-door mechanical continuous traction 15 min, sitting.	C: Physiotherapy scapular strengthening 3 × 10 rep., cranio-cervical strengthening 10 × 10 s	NPRS 0–10 (4 weeks)	86 (32)
Young I.A., 2009	RCT	Georgia	Clinic criteria: pain, numbness to an arm and at least 3/4 positive test of Wainner’s cluster.	I1: Physiotherapy + mechanical intermittent traction-dorsal thrust mid and high, P-A glide, retraction, rotation, lateral glide in ULTT1, P-A glide, strengthening exercise-traction 15 min, 15° cervical flexion, 50 s tension/10 s relaxation.	C: Physiotherapy + sham traction-dorsal thrust mid and high, P-A glide, retraction, rotation, lateral glide in ULTT1, P-A glide, strengthening exercise-sham traction.	NPRS 0–10 (4 weeks)	81 (12)
Moustafa I.M., 2014	RCT	Egypt	Clinic criteria: neurological exams, exacerbated symptoms in flexion, protraction and reduction of symptoms in retraction, extension, side homolateral side-bending and positive Wainner’s cluster (4/4). Imaging: unilateral hernia C5-C6, C6-C7 confirmed by MR and CT.	I1: Physiotherapy, MT + mechanical intermittent traction, 50 s tension/10 s relaxation 24° flexion. I2: Physiotherapy, MT + FCR H reflex traction.	C: Physiotherapy, TM (laser, TENS, STM, Dorsal thrust, cervical flexor and scapular muscle strengthening.	VAS 0–10 (4 weeks)	216 (27)
Aydin, 2012	RCT	Turkey	Clinic criteria: neurological exam and positive neurotension and spurling test. Imaging: MR.	I1: Physiotherapy + mechanical intermittent traction 7 s tension, 5 s relaxation.	C: Physiotherapy, US, hot packs, TENS, isometric exercise for cervical extensor and flexor, stretching.	VAS 0–100 (3 weeks)	27 (/)
Klaber Moffet J.A., 1990	RCT	England	Clinic criteria: typical symptomatology:	I1: Mechanical continuous traction 25° flexion.	C: Sham traction.	NPRS 0–10 (4 weeks)	100 (/)
Jellad, 2009	RCT	Tunisia	Clinic criteria: typical symptomatology. Imaging: involvement of spinal nerve with herniated disc and/or intervertebral disc degeneration by CT or MR.	I1: Physiotherapy + manual intermittent traction 20 repetitions, 20 s tension/10 s relaxation, most pain-free position. I2: Physiotherapy + mechanical continuous traction 2 sessions of 25 min, 10 min rest intra-session, most pain-free position.	C: Physiotherapy US, infrared, massage, cervical spine mobilization, muscle strengthening via isometric contraction of flexor and extensor muscles, stretching exercises.	VAS 0–100 (4 weeks)	39 (/)

Legend: CT= computed tomography imaging; FCR= flexor carpi radialis H-reflex-based traction method; MT=Manual therapy; MR= Magnetic resonance imaging; NPRS= Numeric Pain Rating Scale; P-A= posterior to anterior; RCT=Randomized controlled trial; PEDro= Physiotherapy Evidence Database; PICO= population, intervention, comparison and outcomes; STM= Soft tissue mobilization; TENS= Transcutaneous electrical nerve stimulation; ULTT1= Upper Limb Tension Test 1; US= Ultrasound; VAS= Visual analogue scale; I1= Intervention number 1; I2= Intervention number 2; C= Comparator.

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
