# Peer review of "Traction Therapy for Cervical Radicular Syndrome is Statistically Significant but not Clinically Relevant for Pain Relief. A Systematic Literature Review with Meta-Analysis and Trial Sequential Analysis"

_jcm, 2020, doi:10.3390/jcm9113389_

Round 1
Reviewer 1 Report
Introduction
"There is general consensus on surgical approach": this statement is apodictic and does not refer to the issues of length of radiculopathy, of motor deficits, and bothering of activities. If so there would be no algotithmen including conservative treatment. the Reader should learn of algorithms in terms of CRS. So adopt your statement.
Traction is a passive therapy modality. This should be presented to the reader because this passive therapy modalities have to be compared to spontanous course (sham or no therapy), to other passive modalities and to active modalities (like exercise). The reader should be informed early of those different approaches. So the methods should be differenciated in this point (" traction in addition to other intervention (e.g., physiotherapy) versus other intervention) as possible (other passive / other active intervention).
Discussion
"this treatment option should be reserved for patients with acute symptoms or patients likely to accept the technique but not patients with poor compliance": this statement is confusing because compliance/adherence was not assessed. Moreover polypragmatism and overtreatment should be avoided and not advised.
All studies were performed while spontanous course was working. This limitation - as no study was performed versus no therapy - has to taken into account.
If any modality is added to Standard procedure the patient will receive something more. This gain in supply of care should be discussed as placebo effect or systematic error (bias) in all study protocolls. Randomisation of multimodal procedures should care for balanced therapeutic dose in all treatment arms.
Author Response
Introduction
"There is general consensus on surgical approach": this statement is apodictic and does not refer to the issues of length of radiculopathy, of motor deficits, and bothering of activities. If so there would be no algotithmen including conservative treatment. the Reader should learn of algorithms in terms of CRS. So adopt your statement.
Author Response
We agree and we thank the reviewer for let us the opportunity to better describe the status of available treatments (and related algorithm). We revised the introduction at lines 37 to 49 page 2.
“…There is no clearly established consensus regarding indications for CRS treatments. A common approach recommends surgical treatment when there are several neurological deficits, a strong correlation between imaging and clinical signs and symptoms, failure or relapse of symptoms after 6 weeks of conservative treatment. Guidelines suggest surgery for rapid relief of pain in CRS in short terms when of radicular pain is caused by degenerative cervical spondylosis. Nonetheless, evidence is insufficient for defining the optimal time to surgery and the best surgical approach. In all other conditions, conservative treatment should be preferred. In fact, patients undergoing conservative treatment reported good to excellent outcomes in up to 90 % of cases. Conservative treatment of CRS included different interventions such as manual therapy (mobilization and manipulation), physical therapy and traction…..”
Traction is a passive therapy modality. This should be presented to the reader because this passive therapy modalities have to be compared to spontanous course (sham or no therapy), to other passive modalities and to active modalities (like exercise). The reader should be informed early of those different approaches. So the methods should be differenciated in this point (" traction in addition to other intervention (e.g., physiotherapy) versus other intervention) as possible (other passive / other active intervention).
Author Response
We take the opportunity to better explain the PICO question of this review. We are interest to first explore the comparison of traction therapy (alone or in addition to other intervention) to spontanous course (sham or no therapy). We acknowledge the reviewer for the suggestion about the different approaches that now are added in the last paragraph of the introduction. We amended accondingly the eligibility criteria in the method section in order to better present the question to the reader. Then, we take advantages from reviewer’s suggestion about this important issue addressing this concept in the discussion section as limitation: once the superiority of the traction versus sham or no therapy is clarified, future studies should investigate the head-to-head comparison of interventions (lines 342-344, page 14).
Discussion
"this treatment option should be reserved for patients with acute symptoms or patients likely to accept the technique but not patients with poor compliance": this statement is confusing because compliance/adherence was not assessed. Moreover polypragmatism and overtreatment should be avoided and not advised.
Author Response
We revised the sentence focusing only on the outcome pain and softening the clinical implications related to the interpretation of statistically significant but not clinically relevant results: “The use of traction therapy for CRS does not appear to be essential in clinical practice. However, pain measure is a patient reported outcome and can be influenced by psychological mechanism. In the light of these considerations, this treatment option could only be reserved for patients with acute symptoms or patients that show a reduction of symptoms (i.e., pain) when subjected to cervical traction during the physical examination”. We provided a reference for sustain this statement.
All studies were performed while spontanous course was working. This limitation - as no study was performed versus no therapy - has to taken into account.
Author Response
Only one study investigated traction versus sham traction. However, we take advantages from reviewer suggestions and we addressed this point in the discussion section as limitation (lines 342-344, page 14).
If any modality is added to Standard procedure the patient will receive something more. This gain in supply of care should be discussed as placebo effect or systematic error (bias) in all study protocolls. Randomisation of multimodal procedures should care for balanced therapeutic dose in all treatment arms.
Author Response
We agree and we now better argued this point in the limitations of discussion section (lines 345-358, page 14).
Reviewer 2 Report
This study Colombo et al reviewed randomized controlled trials and quantified using meta-analysis of the effectiveness of traction therapy in patients with cervical radicular syndrome. Perhaps I have misunderstood, but the way the manuscript is written suggests the clinical relevance of the condition being studied is not currently relevant. Here are my comments:
[1] The abstract is verbose, with many technical words thrown in as "run-ons", especially in the very beginning. It is difficult to decipher exactly what the authors are trying to achieve using these analyses. Clear description is necessary.
[2] How important is this condition worldwide and why ? Some statistics are needed to show relevance of why this review is needed now.
[3] In Table 1, Utah is listed as a country, whereas Utah is a state in the USA.
[4] The studies listed in Table 1 show one reference that is from 2019. Other studies date back six years and as far back as 1990. These are very old studies, suggesting that the clinical relevance is outdated. What is the current status of this study with patient data in more recent years?
Author Response
This study Colombo et al reviewed randomized controlled trials and quantified using meta-analysis of the effectiveness of traction therapy in patients with cervical radicular syndrome. Perhaps I have misunderstood, but the way the manuscript is written suggests the clinical relevance of the condition being studied is not currently relevant. Here are my comments:
Author Response
We confirmed the interpreation. With low quality of evidence traction seems to be superior to other conservative treatments alone when combined with other conservative treatments in reducing pain in patients with CRS at 3-4 weeks follow-up assessment but not clinically relevant.
We have the manuscript re-revised twice by a native English speaker (Kenneth Adolfv BRITSCH, Avicenna snc https://aiti.org/it/profilo/kenneth-adolf-britsch). We acknowledged him in the Acknowledgments section in order to be transparent.
[1] The abstract is verbose, with many technical words thrown in as "run-ons", especially in the very beginning. It is difficult to decipher exactly what the authors are trying to achieve using these analyses. Clear description is necessary.
Author Response
We better revised the abstract acconding to the arised questions looking for a better consistency of terms across aim, methods and results. Moreover, we took inspiration from the last published SRs in the JCM having the same format (example, https://www.mdpi.com/2077-0383/9/10/3179)
[2] How important is this condition worldwide and why? Some statistics are needed to show relevance of why this review is needed now.
Author Response
We revised the introduction by providing incidence data. “A more recent study from the US military found an incidence of 1.79 per 1000 person-years”.
We also specify why this review is needed. “Patients undergoing conservative treatment reported good to excellent outcomes in up to 90 % of cases [Saal 1996]”. About these conservative treatments, there is uncertian about the effectiveness of traction since previous SRs reported different methods and findings.
[3] In Table 1, Utah is listed as a country, whereas Utah is a state in the USA.
Author Response
We thank the reviewer for pointing out this oversight and we edit the table 1 accondingly.
[4] The studies listed in Table 1 show one reference that is from 2019. Other studies date back six years and as far back as 1990. These are very old studies, suggesting that the clinical relevance is outdated. What is the current status of this study with patient data in more recent years?
Author Response
We followed the high standards for conduct and report a Systematic Review (for quality AMSTAR II, for reporting PRISMA). We also followed the MECIR (Methodological Expectations of Cochrane Intervention) and Cochrane Handbook guidances for Cohrane SRs (which ares considered the gold standard). “The goal of systematic review searches is to identify all relevant studies on a topic, irrespective of the publication date”. (https://handbook-5-1.cochrane.org/chapter_6/6_4_9_language_date_and_document_format_restrictions.htm)
The clinical relevance depends on the outcome and choice of the outcome measure. The clinical relevance does not depend on the publication date. We included all the studies answering the pico question. We searched up to May 2020. The current status of this study is up-dated.
(Coster WJ. Making the best match: selecting outcome measures for clinical trials and outcome studies. Am J Occup Ther. 2013;67(2):162–70. http://dx.doi.org/10.5014/ajot.2013.006015.Medline:23433270.)
For details, the threshold choice of our clinical relevance becomes from a recent panel consensus meeting on spine. We declared a priori minimal important difference (MID) adopted in the methods “.. a minimal important difference (MID) between groups of 20 points on the outcome pain measured on a standardized scale of 0-100 mm (43) “. Our findings are always expressed in same scale 0-100 mm (mean difference): this can help clinians to direcly read the improvement.
(Ostelo, RWJG, Deyo RA, Stratford P, Waddell G, Croft P, Von Korff M, et al. Interpreting change scores for pain and functional status in low back pain: towards international consensus regarding minimal important change. Spine. 2008 Jan 1;33(1):90–4).
Round 2
Reviewer 1 Report
Thank you for your revisions. You improved your work significantly.